# Attention-Guided Prostate Lesion Localization and Grade Group Classification with Multiple Instance Learning

**Ekaterina Redekop**[1,2]                                    EREDEKOP@G.UCLA.EDU
**Karthik V. Sarma**[1,2]                                        KSARMA@UCLA.EDU
**Adam Kinnaird**[3]                                            ASK@UALBERTA.CA
**Anthony Sisk**[4]                                       ASISK@MEDNET.UCLA.EDU
**Steven S. Raman**[2]                                  SRAMAN@MEDNET.UCLA.EDU
**Leonard S. Marks** [5]                               LMARKS@MEDNET.UCLA.EDU
**William Speier**[1,2]                                          SPEIER@UCLA.EDU
**Corey W. Arnold**[1,2,4]                                    CWARNOLD@UCLA.EDU

[1] *Computational Diagnostics Lab, University of California, Los Angeles, CA 90024, USA*

[2] *Department of Radiology, University of California, Los Angeles, CA 90024, USA*

[3] *Division of Urology, Department of Surgery, University of Alberta*

[4] *Department of Pathology, University of California, Los Angeles, CA 90095, USA*

[5] *Department of Urology, University of California, Los Angeles, CA 90095, USA*

**Editors:** Under Review for MIDL 2022

## Abstract

Lesion localization is a component of prostate magnetic resonance imaging (MRI) evaluation and is essential for targeted biopsy by enabling registration with real-time ultrasound. Most previous work on prostate cancer localization has focused on classification or segmentation assuming the availability of radiology annotations. In this work, we propose to use an unsupervised attention-based multiple instance learning (MIL) method in an application for the classification and localization of clinically significant prostate cancer. We train our model end-to-end with only image-level labels instead of relying on voxel-level annotations. We extend MIL method by operating both on patches and the whole size images to learn local and global features, which improves classification and localization performance. To better leverage the relationships between multi-modal data, we use an architecture with multiple encoding paths, where each path processes one image modality. The model was developed on a dataset containing 986 multiparametric prostate MRIs and achieved $0.75 \pm 0.03$ AUROC using 3-fold cross-validation in prostate cancer Grade Group classification. Lesion localization analysis showed 70-80% sensitivity for GG $\geq 3$ at less than one false positive (FP) per patient and 65% of GG2 at one FP per patient.

**Keywords:** Multiparametric prostate MRI, Multiple instance learning, Image classification prostate cancer.

## 1. Introduction

Prostate cancer (PCa) is the most common internal malignancy in men (Sung et al., 2021). The diagnosis of prostate cancer entails the use of digital rectal examination (DRE) and prostate specific antigen (PSA) testing. For men with palpable lesions and/or high PSA values, biopsy may be indicated and is typically performed transrectally using ultrasound guidance with sampling corresponding to a pre-defined grid in order to uniformly sample the prostate. Pathological grading is used to assess biopsied tissue, with lesion aggressiveness rated via the Gleason score (GS), which is assigned based on a pathologist's visual

review of histological morphology (Epstein et al., 2016). The ISUP Grade Group system has been adopted to further categorize Gleason scores based on risk stratification into five categories (1-5), with increasing risk of cancer mortality corresponding to increasing Grade Group number (GG). Unfortunately, grid-based (i.e., systematic/sextant) biopsies demonstrate clinically significant under-grading of cancer due to incomplete sampling. To improve the diagnostic accuracy of prostate biopsy, multiparametric magnetic resonance imaging (mpMRI) can be used to target areas of the prostate deemed suspicious by a radiologist by fusing MRI data with real-time ultrasound (US) data for needle guidance during the biopsy procedure. MRI has been proven to be the most accurate noninvasive technique for early detection and staging of prostate cancer (Turkbey and Choyke, 2012).

Computer-aided diagnosis (CAD) systems have been developed for prostate cancer detection and localization using mpMRI. With recent advances in deep learning, many CADs are based on convolutional neural networks (CNNs), which have shown promise in detecting cancerous regions on mpMRI (Kiraly et al., 2017; Sumathipala et al., 2018). Cao et al. (Cao et al., 2019b) developed a CNN to jointly detect prostate cancer lesions and accurately segment lesion contours and achieved 75.1% sensitivity at the cost of one false positive (FP) per patient. In other work, Cao et al. proposed a multi-class CNN to jointly detect PCa lesions and predict their aggressiveness using Gleason GG (Cao et al., 2019a). The described framework achieved 87.9% sensitivity at one FP per patient for clinically significant lesion detection task (GG $\geq$ 2) and area under the curve of 0.81 for classification of clinically significant cancer. These works rely on the availability of tumor segmentation masks manually drawn by expert radiologists, which is a time-consuming and subjective task.

The challenges in obtaining voxel-level ground truth masks have led to a growing interest in developing semi- or weakly-supervised approaches. Multiple instance learning (MIL) is a weakly-supervised approach where each labelled sample is represented as a set (or 'bag') of instances. The objective of MIL is then to classify the bag of instances rather than the individual instances. MIL has shown high performance on prostate biopsy whole slide images (WSIs) which have enormous size and are typically divided into smaller patches for analysis based on only slide-level labels (Campanella et al., 2019; Lu et al., 2021). Recently Ilse et al. presented an attention-based MIL model which leverages trainable attention module to visualize the relative contribution of instances for final prediction without sacrificing bag-level prediction performances (Ilse et al., 2018). Attention maps learned with the proposed approach were consistent with cancerous regions identified by pathologists during diagnosis (Li et al., 2021).

The main contributions of this paper are summarized as follows. First, in our work we explore the effectiveness of an MIL approach applied to prostate MRI and compare its performance to a method where attentions are learned based on the whole size images. We argue that dividing images into smaller patches for training an attention-based MIL model provides better localization and reduces the number of false positive lesions. Second, to better leverage the multi-channel nature of MRI data, we explore the influence of early and late fusion strategies to improve lesion classification and localization quality. Third, local and global features are learned in two separate branches of the network to further improve classification and lesion localization with high sensitivity and low per-patient false positive.

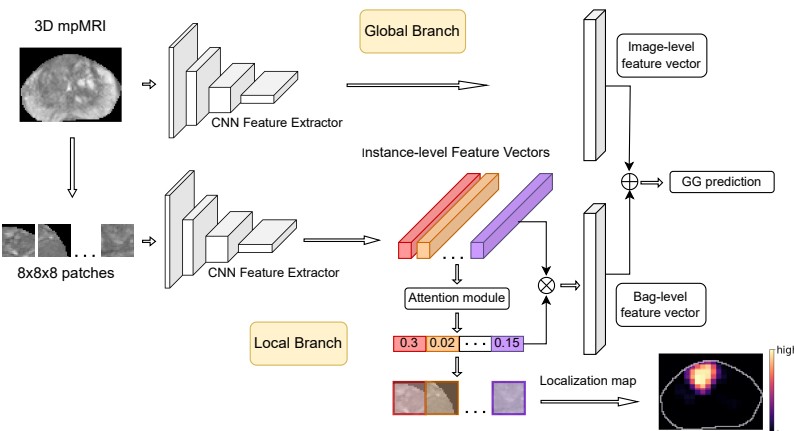

Figure 1: Overview of the proposed prostate mpMRI classification and lesion localization framework. The model consists of two branches, both containing a CNN feature extractor. The local branch is trained on patches in the attention-based MIL framework, while the global branch is trained on whole size images.

## 2. Method

The general pipeline of the proposed approach is presented in Figure 1 and consists of two branches: a global branch that operates on the entire image, and a local branch that operates on image patches (see Section 2.4 for details). Each of the branches can process 3D multiparametric data using different fusion strategies (see Section 2.3 for details). 3D MRI is passed through the global branch to obtain an image-level feature vector. Patches, extracted from the image used to generate the global feature vector, form a bag and the bag is passed through the local branch, which is essentially an attention-based MIL framework (see Section 2.2 for details). Bag-level feature vectors, obtained from the output of the local branch, is added to the image-level vector (output from global branch) and the classification prediction is made. Each patch is associated with a learned attention weight. Given the stride length, each pixel occurs in multiple patches and may thus be associated with different attentions. A localization map can be obtained by averaging these attentions, resulting in pixel-level attention specificity. For comparison, in Section 2.1 we describe alternative attention-based localization methods based on the whole size image analysis.

### 2.1. Attention-guided localization based on the whole image

To locate the discriminative regions for whole size image classification we follow ideas presented in previous work (Guan et al., 2018; Chen et al., 2021). Given an input image, let $F \in \mathbb{R}^{C \times H \times W \times D}$ represent the activation outputs of the last convolutional layer, which are then fed into a global average pooling (GAP) layer, followed by a fully-connected (FC) layer. We denote the weight matrix of the FC layer as $W \in \mathbb{R}^{C \times K}$, where $K$ is the number of classes in the classification model. Finally, for each class $k$ we define an attention map

$A_k \in \mathbb{R}^{\times H \times W \times D}$ as,

$$A_k(x, y, z) = \sum_{c=0}^{C-1} W_{ck} F_{cxyz} \tag{1}$$

The obtained attention heat map $A_k$ is resized to the original image size, and the most discriminative regions $M_k$ of the given image are calculated as

$$M_k(x, y, z) = \begin{cases} 1, & A_k(x, y, z) \geq \tau \\ 0, & if A_k(x, y, z) < \tau, \end{cases} \tag{2}$$

where $\tau$ is a threshold that determines the size of the discriminative region and larger $\tau$ leads to a smaller region and vice versa.

## 2.2. Attention-based MIL

In this work, we utilize an attention-based MIL framework (Ilse et al., 2018) that is designed to learn localization information from a classification neural network. Images are regarded as bags composed of their constituent image patches. In our work, the MIL bags with Grade Group 2 (GG2) or higher are called positive bags and others are negative. The GG2 threshold was selected as these tumors are generally considered to be treatment worthy. We note that in our problem context, some instances (patches) inside each bag represent lesion regions, but that this information is not used during model training. A CNN is utilized to extract the feature embedding $h_k$ of each instance $x_k$, where $h_k \in \mathbb{R}^M$, $M$ is the dimensionality of instance features, $K$ - number of instances in the bag. An embedding for the whole bag can thus be calculated using attention-based MIL pooling as proposed by Ilse et al. (Ilse et al., 2018):

$$Z = \sum_{k=1}^{K} \alpha_k h_k, \tag{3}$$

where

$$\alpha_k = \frac{\exp(w^T \tanh(V h_k^T))}{\sum_{j=1}^{K} \exp(w^T \tanh(V h_j^T))}, \tag{4}$$

$w \in \mathbb{R}^L$, $V \in \mathbb{R}^{L \times M}$ - learnable parameters. Image-level prediction can now be obtained by applying a fully connected layer to the bag-level features. Having an attention value $\alpha_k$ for each instance in the bag, we may combine them into an attention map with size equal to initial image size. The most discriminative regions $M_k$ can be calculated according to (2).

## 2.3. Fusion strategies for multi-modal images

Each MRI channel plays a different diagnostic role in prostate tumor classification and localization. One of the most typical approaches to process multi-channel data is early fusion, where different channels are stacked together and passed through the network as multi-channel image. The main disadvantage of this approach is that it is based on the assumption that relation between different modalities is simple, which is not true (Srivastava et al., 2014). To better learn the multi-channel relationships, a late fusion strategy can be applied, where each modality is merged with others in a deep layer after an independent CNN (Xu et al., 2019).

### 2.4. Local-global attention-based MIL

In order to increase patch-based model performance by adding high-level information, we follow ideas presented Jose et al. (Jose and Oza, 2021) in which the MIL model is a local branch that operates on patches of the image and focuses on finer detail and the classification CNN is a global branch that works on the entire image volume and focuses on high-level information. Feature maps from both branches are added and passed through the final classifier that consists of a fully connected layer.

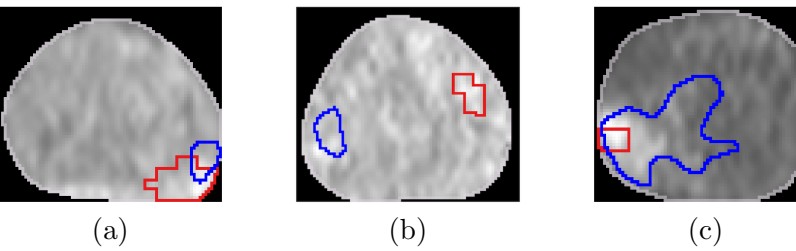

(a)  (b)  (c)

Figure 2: Attention-based lesion localization based on the proposed approach: (a) TP based on IOU and distance criteria; (b) FP based on both criteria; (c) FP based on distance criterion, but TP based on IOU criterion. Localized lesions are indicated with red contour and ground truth lesions with a blue contour.

## 3. Experiment

### 3.1. Dataset

Our dataset contains 986 studies collected from patients who underwent transrectal ultrasound - MRI fusion biopsy (TRUS biopsy) using the Artemis guided biopsy system (Eigen Systems) between 2010 and 2018 at our institution using a standardized protocol and 3T magnet (Trio, Verio, or Skyra, Siemens Healthcare). As part of this clinical process, a radiologist contours a prostate and any regions of interest (ROIs) for targeted biopsy sampling. 3D T2-weighted (T2W) images, apparent diffusion coefficient (ADC) maps, high b-value diffusion weighted images (DWI), and prostate and lesion contour sets were available for this study. MRI preprocessing included bias field correction and interquartile range (IQR)-based intra-image normalization to address the relative nature of MRI intensity values. Each image was normalized to the image-level IQR calculated inside the prostate gland and then values were clipped between two IQRs below the first quartile and five IQRs above the third quartile in order to eliminate outlying values created by imaging artifacts. All images in the dataset have the same voxel spacing and therefore no resampling was needed. We then extracted patches of size 8x8x8 from the grid with 50% overlap. Patches that contained less than 50% prostate gland were discarded from analysis.

In this work, we assume that a prostate mask is available for each MRI (see Figure 4 in the Appendix). Each image is then cropped to the prostate region to reduce input size and improve model convergence. If the gland volume masks are not provided initially, they

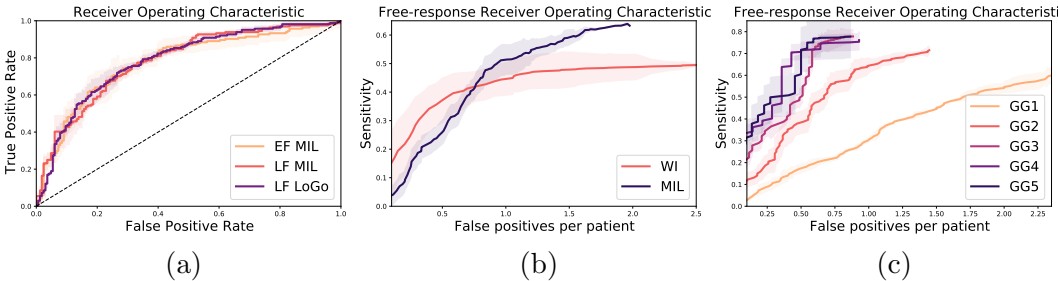

Figure 3: (a) ROC analysis for Gleason Grade Group classification. (b) FROC analysis for WI and MIL comparison. (c) FROC analysis for attention-based lesion detection sensitivity for each specific Gleason Grade Group. The transparent areas are 95% confidence intervals estimated by two times of the standard deviation

can be obtained using existing prostate segmentation models, which have achieved high performance (overall Dice 0.916) (Sarma et al., 2021).

We randomly divided the dataset into 70% for training and 30% for validation, stratifying by patient-level GG determined by the highest GG in each patient's corresponding set of biopsy cores. Overall, dataset consisted of 492 GG1 lesions, 264 GG2 lesions, 110 GG3 lesions, 55 GG4 lesions and 65 GG5 lesions.

### 3.2. Implementation details

We used a 3D version of the LeNet5 model (LeCun et al., 1998) as a backbone for the feature extractor in both the MIL model (local branch) and the whole image classifier (global branch).Taking into account the size of the input patches, we adapted the parameters of convolution and pooling layers (see Table 2 and Table 3 in the Appendix for detailed model architectures). To accommodate the anisotropic nature of our data, the first 3D convolution uses a kernel size of 1 (what makes them effectively 2D convolutions) in the out of plane axis to prevent aggregation of information across distant slices. Sizes of input patches to the MIL model are $8 \times 8 \times 8$, therefore the sizes of feature maps from the last convolutional layer are $64 \times 4 \times 2 \times 2$. Feature maps were flattened and fed into a FC layer with 256 nodes to produce a $M \times 256$ instance embedding matrix, which was forwarded into the attention module (4). The attention module generated an $K \times n$ attention matrix, where $K$ stands for the number of instances in the bag and $n$ stands for number of classes and is equal to one in our case. The instance embedding matrix with size $K \times 256$ can be multiplied by the $K \times 1$ attention vector, which results in $1 \times 256$ bag-level representation that is forwarded into a final classifier consisting of a FC layer.

The initial learning rate was set at $1 \times 10^{-4}$ and was decreased by a factor of 10 if the validation loss did not improve for the last 5 epochs. We used the Adam optimizer and a batch size of one. Random flipping, rotation, and elastic transformations were utilized for data augmentation.

### 3.3. Evaluation metrics

False positive tumor detection is an important metric in prostate cancer diagnosis as they can result in unnecessary biopsies. Thus, a primary goal is to reduce the number of false positive predictions while maintaining sufficient sensitivity. Previous work has found that true positive detections are localized points in or within 5mm of a lesion's center since PCa lesion diameters on corresponding surgical pathology specimens are roughly 10mm larger than the suspicious region in MRI (Litjens et al., 2014). At the same time, as mentioned by Litjens et al., false positives can also arise from large lesions where the distance between the center of the radiologist contour and the predicated lesion exceeds a pre-determined criterion. In this study, the distance criterion was defined as 5mm based on previous literature. When the ground truth lesion is quite large, the distance between the center of mass of predicted and ground truth lesions can be more than 10mm. We observe the same phenomenon in our data, where a localized lesion can be clearly inside the lesion drawn by a radiologist on several 2D slices, but its center of mass calculated using the entire volume is displaced along the y-axis (see Figure 2(c)). We found that intersection over union (IOU) calculated between detected and ground truth lesions is resistant to such cases. The detection is a true positive if it has intersection with one of the ground truth lesions and IOU is higher than 0.01, e.g., IOU is equal to 0.015 for localization result on Figure 2(a). False positive are all detections that are not true positive (see Figure 2(b)).

## 4. Results

We used the area under the receiver operating characteristic curve AUROC and average precision AP computed from ROC and precision and recall (PR) curves, respectively. Figure 3 (a) shows the ROC analysis for GG $\geq$ 2 vs. GG $<$ 2 classification. We compared the performance of the MIL framework with early fusion (EF MIL) and late fusion (LF MIL) using only the local branch (see Section 2.3 for details) for prostate Grade Group classification. Three-fold cross validation results are presented in Table 1. Late fusion MIL showed slightly better performance than early fusion in both AUC and AP values. We therefore used LF for both branches of the local-global approach (LF LoGo), which allows to further increase both metrics and achieve $0.75 \pm 0.03$ AUC. Attention-guided localization based on the whole image (WI) analysis showed lower performance than each of three MIL experiments. We also observe that the WI approach results in less localized lesions (see Figure 5 in the Appendix.)

Lesion localization performance is evaluated using free-response receiver operating characteristics (FROC) analysis. FROC measures the lesion detection sensitivity versus the number of false positive per patient. Comparison of FROC curves for the best-performing MIL and WI models shows that the WI approach achieves higher localization sensitivity when per patient FP is less than 0.7. However, the corresponding sensitivity is less than 0.4, minimizing potential clinical. The overall maximum achieved sensitivity of the MIL framework is higher than the WI approach and is equal to 64% at two FP per patient. FROC analysis for the MIL framework for each specific GG is shown in Figure 3 (c). The model detects 70%-80% of GG $\geq$ 3 lesions at less than one false positive per patient and 65% of GG2 at one FP per patient. For GG1 lesions, sensitivity is low even when FP lesions are largest. This finding is explained by the fact that this particular group contains not only

GG1, but also negative-biopsy lesions (i.e., lesions outlined by the radiologist that produced negative biopsies). Visualization of successful and failed localization cases for each Gleason Grade Group is shown in Figure 5 in the Appendix.

Direct comparison with previous work on prostate mpMRI Gleason Grade Group classification is not possible because the used datasets are different. However, 75.1% sensitivity at one FP per patient reported in the work by Cao et al. (Cao et al., 2019b) was achieved under the assumption that at least one lesion with GG $\geq$ 2 was identified in histologic examinations for each volume. In the other work, Cao et al. reported 87.9% sensitivity at one FP for clinically significant lesions (GG $\geq$ 2), but for all lesions the achieved sensitivity was slightly higher than 60% at one FP (Cao et al., 2019a). In our work we achieved 77.4% sensitivity at less than one FP for clinically significant lesions and 60% at 1.5 FP for all lesions. Unlike other methods, lesion masks provided by radiologists were not used in training our model.

Table 1: Model performance of prostate GG $\geq$ 2 vs. GG $<$ 2 classification. EF MIL and LF MIL - feature extraction in local branch only (MIL framework) using early and late fusion strategies correspondingly. LF LoGo - feature extraction in both local and global branches with late fusion strategy. WI - attention-guided localization based on the whole image. AUROC - area under the receiver operating characteristic curve, AP - average precision.

|  | EF MIL | LF MIL | LF LoGo | WI |
|---|---|---|---|---|
| AUROC | $0.73 \pm 0.04$ | $0.74 \pm 0.05$ | $0.75 \pm 0.03$ | $0.73 \pm 0.03$ |
| AP | $0.73 \pm 0.04$ | $0.74 \pm 0.05$ | $0.75 \pm 0.03$ | $0.71 \pm 0.02$ |

## 5. Conclusion

In this work we applied an attention-based MIL framework to the task of prostate cancer Grade Group classification with lesion localization. We found that a late fusion strategy better leveraged relationships between data modalities and resulted in higher performance compared to early fusion. Extending the model to operate on both patches and the whole image using two separate network branches in combination with late fusion strategy showed the best performance and produced $0.75 \pm 0.03$ AUC in classification of GG $\geq$ 2 vs. GG $<$ 2. FROC analysis shows that the model is able to localize cancerous regions with GG $\geq$ 2 with 77.4% sensitivity at less than one false positive per patient. Our model outperforms attention-guided localization based on the whole image according to both ROC and FROC analysis.

Our model is trained with weak supervision using only image-level labels with lesion masks used only for visualization and the calculation of false positives. This paradigm allows us to easily accommodate clinically-generated data and is thus more general than approaches requiring pixel level annotations. A limitation of our work, and any work that includes patients with only biopsy results, is that we do not have surgical resection specimens to provide a ground truth.

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

## 6. Appendix

Table 2: Local branch architecture. Input shape $K \times 3 \times 8 \times 8 \times 8$.

| Layer | Type | Output shape |
|---|---|---|
| 1 | conv((1,3,3),1,0)-32+ReLU | $K$ x 32 x 8 x 6 x 6 |
| 2 | maxpool(2, 1) | $K$ x 32 x 7 x 5 x 5 |
| 3 | conv((3,3,3),1,0)-64+ReLU | $K$ x 64 x 5 x 3 x 3 |
| 4 | maxpool(2, 1) | $K$ x 64 x 4 x 2 x 2 |
| 5 | fc-256 + ReLU | $K$ x 256 |
| 7 | mil-attention-128 | $K$ x 1 |
| 8 | fc-1 + sigm | 1 |

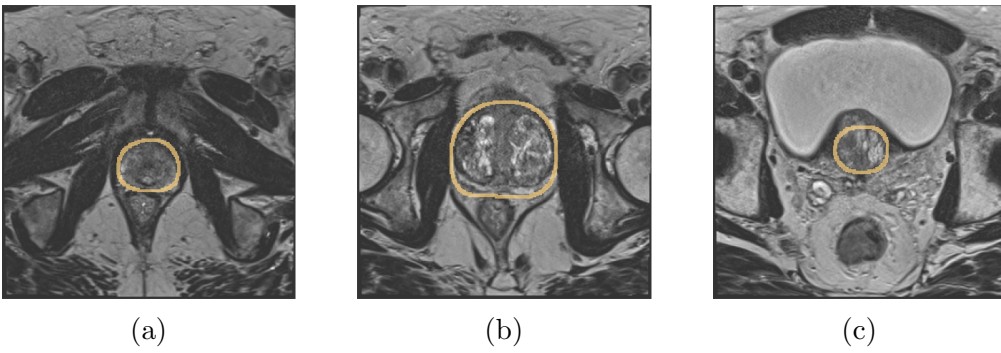

(a)            (b)            (c)

Figure 4: Example slices from a dataset sample (T2W) with parts of the prostate: (a) apex (lower part) (b) midgland (c) and base (upper part)

Table 3: Global branch architecture. Input shape $1 \times 3 \times 32 \times 64 \times 64$ (median shape of the cropped prostate region).

| Layer | Type | Output shape |
|---|---|---|
| 1 | conv((1,3,3),1,0)-32+ReLU | 1 x 32 x 32 x 62 x 62 |
| 2 | maxpool(2, 2) | 1 x 32 x 16 x 31 x 31 |
| 3 | conv((3,3,3),1,0)-64+ReLU | 1 x 64 x 14 x 29 x 29 |
| 4 | maxpool(2, 2) | 1 x 64 x 7 x 14 x 14 |
| 5 | conv((3,3,3),1,0)-128+ReLU | 1 x 128 x 5 x 12 x 12 |
| 6 | maxpool(2, 2) | 1 x 128 x 2 x 6 x 6 |
| 7 | fc-256 + ReLU | 1 x 256 |
| 8 | fc-1 + sigm | 1 |

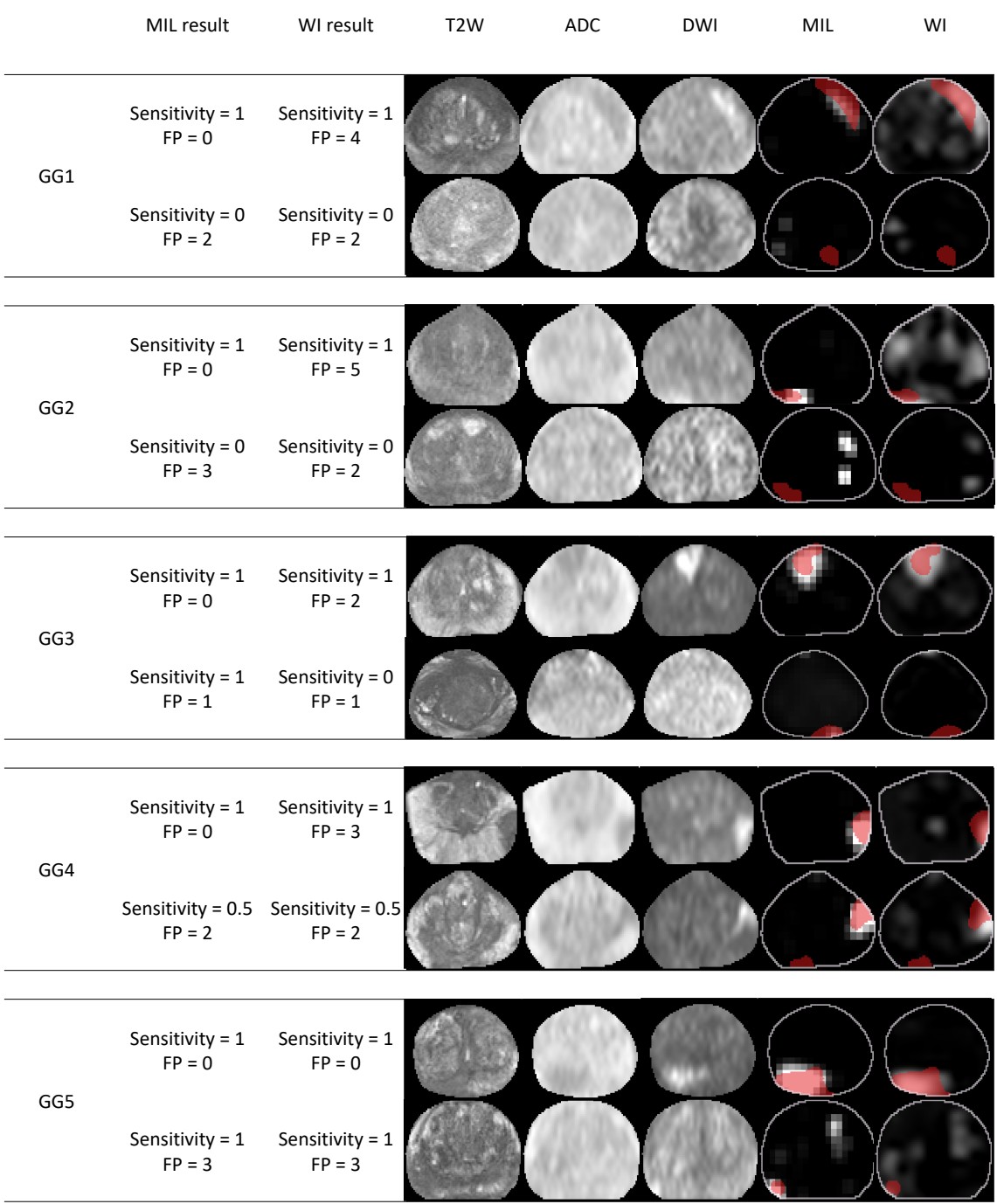

Figure 5: Localization maps obtained by proposed approach (MIL) and whole image based approach (WI) for different GG and different Sensitivity/False Positives number.

**Additional experiments** Accuracies on the individual Gleason Grade Groups (GG) are presented in Table 4. The model is least accurate for samples from class '1' (GG $\geq$ 2) which were assigned with GG2. This can be explained by the fact that dominant pattern (Gleason 3) in GG2 (Gleason 3+4) is the same as GG1 (Gleason 3+3).

Table 4: Accuracy evaluation on the individual Gleason Grade Group.

|     |                       | GG1 | GG2 | GG3 | GG4 | GG5 |
| --- | --------------------- | --- | --- | --- | --- | --- |
|     | mean samples number   | 164 | 88  | 36  | 18  | 22  |
| Acc | EF MIL                | $0.7 \pm 0.05$  | $0.6 \pm 0.06$ | $0.72 \pm 0.1$  | $0.67 \pm 0.01$ | $0.84 \pm 0.03$ |
|     | LF MM                 | $0.71 \pm 0.07$ | $0.6 \pm 0.02$ | $0.7 \pm 0.1$   | $0.68 \pm 0.02$ | $0.84 \pm 0.03$ |
|     | LF LoGo               | $0.71 \pm 0.05$ | $0.6 \pm 0.05$ | $0.72 \pm 0.09$ | $0.7 \pm 0.01$  | $0.84 \pm 0.02$ |

