# OpenReview forum: "Attention-Guided Prostate Lesion Localization and Grade Group Classification with Multiple Instance Learning"
_MIDL.io/2022/Conference — MIDL 2022_

### Official Review · Reviewer_k64e · 2022-01-21

**Confidence:** 4
**Preliminary Rating:** 4
**Recommendation:** Oral

**Summary:**

The paper evaluates a previously published MIL approach using simple CNNs with gated attention for Gleason grading of prostate mpMRI images and compares it against a different attention-guided lesion localization that does not use a bag of patches for MIL.  It also compares early and late fusion strategies for the different MRI channels, and adopts an idea from another recent paper integrating local MIL features from patches with global image features. The methods all make use of (and therefore require) a prostate ROI mask, and the dataset consists of nearly 1000 studies from US/MR-fusion-guided biopsy patients with Gleason scores and lesion contours (not used for training). The approach probably benefits from the fact that the cropped prostate ROIs are small enough and thus may not translate well to many other medical image segmentation problems in which the input volumes are much larger.

**Strengths:**

The paper picks a number of recent approaches and gives justifications for their use and combination.  The method is well suited for the problem and the research questions are relevant.  The problem of mpMRI-based Gleason grading is relevant and interesting as well, and the dataset of nearly one thousand cases, fairly balanced when considering the well-justified (but of course debatable) threshold of GG score 2, appears to be sufficient for evaluation of the method and for addressing the research questions.

**Weaknesses:**

I find that some details are missing (I will give examples below), which makes it harder to follow and probably impossible to reproduce the paper (assuming one would have a comparable dataset).  Given the length limit and complexity of the paper, it is understandable that not all evaluation results can be reproduced, but I find it hard to draw the same conclusions from the results included in the paper as the authors.  No statistical tests are used.  Overall, I have the impression that the results of all compared methods are good (and interesting to the reader), but the claimed differences are small and sometimes debatable.

**Deanonymize Review:**

yes

**Detailed Comments:**

Missing details:
1) How is the global feature vector computed? Via global pooling? The referenced Medical Transformer paper uses feature maps for segmentation, so that is different from the GG classification problem at hand.
2) LeNet-5 uses unpadded convolutions and therefore reduces the image size much more than the 8x8x8 patch sizes would allow. Hence, I assume that padded convolutions were used.
3) However, even with padded convolutions, the feature maps of the last convolutional layer would be 64x4x4x4, not 64x4x2x2, as stated by the authors. Except if the second pooling layer was meant, in which case one would get 64x2x2x2. I assume that the authors have also restricted the first pooling layer to 2D, but that should be mentiond.

Conclusions from results:
4) The differences between early and late fusion in AUC and AP values are small and may not be statistically significant. In particular, since the ROC curves do seem to cross a few times, I wonder if the comparison would look the same when restricting the area to the interesting ranges of operating points. (The paper argues that the number of FP should be kept low.)
5) "Comparison of FROC curves for the best-performing MIL and WI models shows that MIL is able to achieve higher sensitivity at the same number of per patient false positive" -> from Figure 3b, I find that this holds for FP > 0.7 per patient, but the opposite is true for lower FP.  Since the next sentence explicitly gives example values for interesting FP values to look at (FP < 1 and FP = 1), it looks as if the operating range < 0.7 FP is not uninteresting, so I find the observation strangely one-sided. (Again, the results for the MIL approach look good and interesting, I just find these conclusions incomplete.)  Considering the shaded confidence intervals would further strengthen this critique.

I find it very curious that the LeNet-5 model was chosen for the architecture (although it was obviously adapted to 3D data, so it is not exactly the old architecture); I have not seen this being used in any other recent publication.  On the other hand, it is probably not ill-suited for this purpose, even though modern tooling usually focuses on and optimizes 3x3 convolution kernels.

8x8x8 patches are really small, I wonder how complex the features can be at that level. Maybe it would be possible to process the full volume (or use valid convolutions with overlapping tile strategy) and to do the MIL patching on the extracted feature maps instead of on the input image, so that the receptive field is larger than 8x8x8.

"a global branch that works on the original resolution of the image" maybe should be "original extent"? Apparently, the resolution (/voxelsize) is not changed anywhere in this work, or I have missed it.

The description of the training strategy is sufficiently clear and looks appropriate.  Turning the Gleason grading into a binary MIL classification problem by introducing score thresholds is a good approach for this application, as far as I can see.

Small mistakes:
- The figure references in 3.3 are wrong; it took me a while to find that probably Figure 2 is meant, not Figure 4 (which is even hyperlinked).
- "lower performance than" (not "then")

**Final Rating After The Rebuttal:**

5: Strong Accept

**Justification Of The Final Rating:**

If there was a "moderate accept" rating, I would give this. While there are some weaknesses, it is overall an interesting and relevant work which I assume to be of interest to a significant part of the MIDL audience. The authors have also answered to some of my most important criticism, e.g., pointing out that the interesting operation point is somehow between 0.7 and 1 FP per case. (I hope this is more clear now in the revised paper as well; I cannot read it again today.)

**Paper Type:**

both

**Questions To Address In The Rebuttal:**

I would like the two main topic areas above to be addressed:
1) The missing architectural setup details should be added to the paper.
2) The conclusions drawn from the results should IMO be changed / phrased differently, or the rebuttal should provide convincing counterarguments that justify the comparative statements as they currently are.

**Special Issue:**

no

---

### Official Review · Reviewer_97rP · 2022-01-25

**Confidence:** 4
**Preliminary Rating:** 4
**Recommendation:** Poster

**Summary:**

The paper describes a methodology for the localisation/classification of prostate cancer in MRI data. The methodology is based on an unsupervised attention-based multiple instance learning approach, which takes both local and global aspects into account. The evaluation shows promising initial results.

**Strengths:**

This is an image-based label deep learning approach in prostate localistaion/classification.

The methodology takes both local and global aspects into account.

The methodology takes multi-modal data into account.

**Weaknesses:**

The translation to other application areas could have been described in more detail.

The developed methodology seems to be based on a set of existing components, which makes the novelty of the developed methodology less clear.

**Deanonymize Review:**

no

**Detailed Comments:**

See below.

There are references in the text to non-existing tables.

**Paper Type:**

both

**Questions To Address In The Rebuttal:**

To what extent is the developed methodology novel.

How useful is the developed methodology in other application areas.

A full evaluation on the individual Gleason grades would be of interest.

Where doe sthe developed approach go wrong.

Are the differences significant.

**Special Issue:**

no

---

### Official Review · Reviewer_16ep · 2022-01-25

**Confidence:** 4
**Preliminary Rating:** 2
**Recommendation:** Poster

**Summary:**

The paper presents an unsupervised attention-based multiple instance learning method for classification and localization of prostate cancer. Experiments were performed on a private dataset of about 1000 studies indicating that the presented method could be beneficial (at least in comparison to other variants presented). The code is not available.

**Strengths:**

- The authors provide a detailed and understandable introduction to the medical background.
- The authors tackle the important problem of prostate lesion localization and GG classification.
- The results suggest that the presented approach could be beneficial


**Weaknesses:**

-Overall, the paper reads very well, but I found it difficult to find the individual steps in the methods section in the overall picture. Furthermore, I missed some information to better understand the method and the results (e.g. how exactly does the localization result look like).

-Moreover, due to missing and wrong references etc. the work is in parts hard to follow.

-The discussion of the results, especially in the context of the work presented in the Related Work section, is very brief.

-The authors perform some experiments to evaluate the performance of their methods, however, they do not compare it with methods of other groups.


**Deanonymize Review:**

no

**Detailed Comments:**

-  Figure 1 is not referred to. Please add in each subsection a reference to the corresponding part visualized in Fig 1. That makes it easier to follow and understand the method.
-  “Cao et al. (Cao et al., 2019) developed a CNN to jointly detect prostate cancer lesions and accurately segment lesion contours and achieved 75.1% sensitivity at the cost of one false positive per patient. ”  In this cited work, a dataset for bone age is used. The word “prostate” is not mentioned once in that paper. Is that the correct reference?
-  In the overview figure, it is not clear to me where the localization output is generated. The last arrow indicated the GG prediction but no localization prediction is mentioned. Please clarify and adapt the figure.
- In Figure 2, you mention the “distance criterion”, however, it is not properly defined and explained in the paper.
-Figure 2 indicated that “Localization” equals a segmentation results without the need for an overall consistent segmentation result. Is that correct? Please highlight more what exactly is predicted by your network for the localization part.
- For a reader not familiar with prostate MRI images, it would help to have one figure showing a larger field of view around the prostate to get a better feeling/overview.
- “Three-fold cross validation results are presented in Table 4. ” There is not Table 4. It seems that a wrong reference was used.
- In Table 1, what does LoGo mean (it is not mentioned in the text at all)? Moreover, it is helpful to add a more detailed description including the names of the abbreviations so that the reader doesn’t have to search for it in the text.
- Please add the comma in “In this work, we..” (abstract, sec 2.2., 3.1, ..)
- Please do not use abbreviations without introducing them (e.g. AP value)


Questions:
- “Some work has studied the ability of a network to perform both detection and classification of clinically significant prostate cancer at the same time using CNN with two output channels (Kiraly et al., 2017). These works rely on the availability of tumor segmentation masks manually drawn by expert radiologists, which is a time-consuming and subjective task.” If a manual mask is required, why do such methods perform a detection, because you already know where the segmentation mask is located?
- Did you finetune some hyperparameters during training (e.g. best model selection)? How is that done? I am wondering because you only split the dataset in training and validation and do not have any further test dataset. Often the validation dataset is used for model selection etc. and the actual evaluation is performed on the test dataset.
- If the localization output (the contour) is used as a segmentation mask – how good is the segmentation overlap between it and the manually annotated segmentation mask (e.g. Dice)?


**Final Rating After The Rebuttal:**

4: Weak Accept

**Justification Of The Final Rating:**

Thank you for the answer.

I appreciate the updates the authors made. They make it easier to follow and strengthen the work.

I am happy to recommend accepting the paper. I am looking forward to the presentation at MIDL.


**Paper Type:**

both

**Questions To Address In The Rebuttal:**

-Please revise the methods section so that it is more comprehensible (see Detailed Comments) and add more detailed figure captions and introduce abbreviations to make it easier for the reader to follow your paper.
-Please expand the discussion and especially address other methods, limitations of your method and failure cases. It would be helpful to show some examples where your method doesn’t work well and discuss why this might be the case.
-Address the points explained in “Detailed Comments” to improve the quality of your work.


**Special Issue:**

no

---

### Meta-Review · Area_Chair_2xnK · 2022-02-18

**Recommendation:** Accept (Poster)
**Confidence:** 4

**Metareview:**

This paper combines attention, mutliple instance learning and global / patch based approach to localize and classify prostate cancer lesions in MRI. The methodology is well described and suited to the problem at hand, and well validated with a relatively large dataset. The authors also clarified several important aspects raised by the reviewers during the rebuttal phase. I recommend the acceptance of the paper.

---

### Decision · Program_Chairs · 2022-02-28

Accept